# Enabling Trust and Security in Digital Twin Management: A Blockchain-Based Approach with Ethereum and IPFS

**DOI:** 10.3390/s23146641

**Published:** 2023-07-24

**Authors:** Austine Onwubiko, Raman Singh , Shahid Awan , Zeeshan Pervez, Naeem Ramzan 

**Affiliations:** School of Computing, Engineering and Physical Sciences, University of the West of Scotland, Paisley PA1 2BE, UK; austine.onwubiko@uws.ac.uk (A.O.); raman.singh@uws.ac.uk (R.S.); shahid.awan@uws.ac.uk (S.A.); naeem.ramzan@uws.ac.uk (N.R.)

**Keywords:** digital twin, blockchain, Industry 5.0, Ethereum, IPFS

## Abstract

The emergence of Industry 5.0 has highlighted the significance of information usage, processing, and data analysis when maintaining physical assets. This has enabled the creation of the Digital Twin (DT). Information about an asset is generated and consumed during its entire life cycle. The main goal of DT is to connect and represent physical assets as close to reality as possible virtually. Unfortunately, the lack of security and trust among DT participants remains a problem as a result of data sharing. This issue cannot be resolved with a central authority when dealing with large organisations. Blockchain technology has been proposed as a solution for DT information sharing and security challenges. This paper proposes a Blockchain-based solution for digital twin using Ethereum blockchain with performance and cost analysis. This solution employs a smart contract for information management and access control for stakeholders of the digital twin, which is secure and tamper-proof. This implementation is based on Ethereum and IPFS. We use IPFS storage servers to store stakeholders’ details and manage information. A real-world use-case of a production line of a smartphone, where a conveyor belt is used to carry different parts, is presented to demonstrate the proposed system. The performance evaluation of our proposed system shows that it is secure and achieves performance improvement when compared with other methods. The comparison of results with state-of-the-art methods showed that the proposed system consumed fewer resources in a transaction cost, with an 8% decrease. The execution cost increased by 10%, but the cost of ether was 93% less than the existing methods.

## 1. Introduction

Digital twins are used by various organisations in the Industry 5.0 to digitally represent physical assets. In 2023 [1], it has been anticipated that the DT market value will hit $15.66 billion and half of the large industrial companies will adopt the concepts of DTs. The recent development in technology, such as the internet of things (IoT), is used to collect various data generated in the production, management, monitoring, and processing of a product’s entire life cycle. These data are then stored in the cloud. Product life cycle management (PLM) [2] is an effective way a company manages its business activities across products’ life cycle. The collaboration among DT participants is a key factor in completing production on time and within the allocated budget. The fragmentation in production among different participants of DTs, who work together toward a common goal, is geographically spread, which causes delays in data communications among the participants, thereby compromising accountability in information sharing.

A digital twin is a concept that involves virtually representing physical assets or systems. The idea is to utilise sensors to generate real-time information from these physical assets and apply it to their virtual counterparts. However, due to the lack of convergence between the physical and virtual realms, data are often fragmented and isolated. By combining the digital and physical components through data obtained from sensory devices, simulation models can be constructed, enabling proactive maintenance in the context of DT implementation. Any data in the DT are traceable, tamper-proof, and should be transparent without any potential manipulation, as DT technology supports information sharing among different participants [3]. Various digital technologies are employed to virtualise physical assets [4] in order to create trusted virtual models. These models rely on real-time data from the physical assets to generate their real-time conditions and dynamics. This offers a distinctive approach to representing physical assets in the digital form, taking into account their shape, conditions, and status.

With different participants contributing to DT towards a common goal (i.e., production or monitoring an asset) and to manage DT data, the participants involved need access to the DT for information. Consider the participants involved in an industrial phone manufacturing factory, where the conveyor belt is used for phone manufacturing in the factory and relevant data of the production are integrated by the DT. The manufacturer of this conveyor motor should not have access to the production status but rather receive feedback from the maintainer whenever the motor needs repair in order to ensure continuity of the phone assembly line. The quality and process managers should only have access to the current status of the production process, and the maintainer should have access to the motor’s current status.

The issue of information sharing among different participants in the digital twin remains a problem in allowing transparency and traceability among various participants involved in the DT and allowing immutability of information [3]; thus, the data management of a DT, i.e, data storage, security and, sharing, has not been thoroughly recognised. As a result of this, confidentiality and access control issues arise, and these issues pose a significant challenge with a central authority when dealing with a large organisation [5].

Blockchain as an emerging technology is a peer-to-peer connection that has a distributed storage with an encrypted algorithm and consensus protocol to help facilitate how information can be stored and shared among different participants in the organisation that have different incentives in the DT system [6]. Blockchain creates a decentralised solution among the participants of the DT to validate and consent to the integrity of the transaction, as information is shared securely and transparently among participants. Blockchain supports the execution of scripts to define tamper-proof transaction logic called smart contracts, which help ensure unchangeable and transparent digital records of transactions [7]. Blockchain and its smart contract offer various characteristics to support the security of information sharing without requiring a central point of storage.

Some existing work on information sharing in DT has been implemented, but these have limitations, such as cloud storage, and are vulnerable to attacks/security threats such as a man-in-the-middle attack (MITM). Participants in the DT should avoid storing confidential information on a public blockchain, as the use of a public blockchain is a concern for confidentiality since on-chain data are not encrypted [5].

The use of blockchain and DT has been proposed by different researchers [1,3,5]. Ref. [5] proposed a blockchain-based solution for information management that is suitable for DT data sharing. The state-of-the-art includes DT components with access control, providing scalability for sensor data sharing. This approach is demonstrated using a DApp prototype implementation called Ethertwin, which employs the use of Ethereum blockchain and Swarm DHT that evaluates the performance and cost measurement. This is used to achieve information management and sharing with DTs.

Ref. [1] proposed a blockchain-based creation process of DTs to guarantee secure and trusted traceability, accessibility, and immutability of transactions, logs, and data provenance. This approach uses smart contracts to govern and track transactions initiated by the participants involved in the creation of DTs and also uses the InterPlanetary File System (IPFS), a decentralised storage system to store and share DT data. This is used to achieve the creation process of DTs and provide security and cost analysis. Ref. [3] proposed an integrated DT and blockchain framework that can selectively store and share important project-related information traceability. This approach uses Azure as a cloud service platform, which provides blockchain service such as cloud storage and a blockchain network. Azure blockchain uses the Ethereum-based Quorum ledger protocol, designed for high-speed processing of private transactions for authorised participants of the DT. This is used to achieve information sharing across participants and traceable data transactions.

In this paper, we propose a blockchain-based approach, leveraging Ethereum and IPFS to enable trust and security in digital twins. The algorithm we present ensures the integrity and traceability of documents associated with each digital twin. When a document is uploaded, it is stored in IPFS, a decentralised file storage system. The cryptographic hash of the document is then stored on the Ethereum blockchain, allowing for verification of document integrity and providing a transparent record of its history. Multiple participants can own and interact with the digital twin system, with ownership tracked through Ethereum addresses. This algorithm enhances the security and reliability of digital twins.

This paper aims to propose a decentralised information management system that utilises blockchain technology. This proposed system uses a decentralised database, which is secure and has no single point of failure, and nodes do not need to trust each other. The proposed system uses a private blockchain with the integration of IPFS, an off-chain storage that ensures confidentiality. We have demonstrated and tested the proposed system on a real-life phone production use-case.

To achieve information management and sharing, in this paper, we aim to develop and test an integrated blockchain and digital twin, which allow for the sharing and storing of important information about the digital twin to achieve traceability, transparency, integrity, and immutability. The main contribution of this work can be summarised as follows:We propose and design the integration of blockchain and IPFS for efficient data management.We carry out security and cost analysis for the proposed system—a blockchain-based solution for DT using Ethereum.We present smart contracts to govern and control the transactions performed in the proposed system and smart contracts with access control for data and information sharing among the participants of the DT.InterPlanetary File System (IPFS), an off-chain storage of the Ethereum Virtual Machine (EVM), will be integrated to store and share information of the DT.

The rest of this work is organised as follows: Section 2 is the background of the research about blockchain and the digital twin. Section 3 introduces the related works. The structure of the proposed system is laid out in Section 4. Section 5 discusses a case scenario about the research work. Section 6 will discuss the evaluation and analysis of the prototype. Finally, a discussion and conclusion will be drawn in Section 7 and Section 8.

## 2. Background

The background of this work is divided into two sections, Section 2.1 and Section 2.2, which describe the foundation of digital twin and blockchain applications.

### 2.1. Digital Twin

Most researchers have defined the digital twin as virtually representing a physical asset in real time through sensors and other data sources to help mirror the life cycle of the operation of the physical asset and to accurately predict and diagnose any fault or abnormal data [8]. Digital twin was first proposed by Professor Grieves in 2003 [9] for the management of product life cycles, which consists of three parts: physical, virtual, and the data transmission between physical and virtual parts. The idea of the digital twin was proposed as a mirroring model simulation for vehicles and systems and studied in the aerospace field [3] for the product life cycle management by Grieves in the Michigan Executive Course at the University of Michigan in 2003 [10]. In 2010, NASA developed their first DT for a spacecraft, as DT made a significant contribution to the project, and gradually the investigation of DT became a hot trend in aerospace and other industries of the digital world.

There are three main elements of the digital twin: the physical, the virtual, and the data communication between the physical and the virtual element, as shown in Figure 1. The physical element of the DT is the foundation of DT [4]. It creates the virtual model of the physical element to help simulate physical systems and understand their behaviours. The virtual element of the DT is the exact replica of the physical element, which represents the properties and behaviours of the physical element. Communication between the physical and virtual elements of the DT enables information and data exchange for advanced simulation, monitoring, operation, and analysis.

The digital twin has become a vital topic when it comes to communication and information technology [3]. It has been applied in different fields of study such as information management [5,9], transportation [11,12], manufacturing [6,10,13], healthcare and the medical system [14,15], agriculture [16,17], and construction [3]. When designing a digital twin platform, it is essential to prioritise the protection of important data from viruses and hacking attacks due to the utilisation of this information in IoT and cloud computing. With the assistance of blockchain technology, data can now be securely transferred over the internet. Blockchain offers the advantages of security, immutability, transparency, and integrity, thereby facilitating effective monitoring of the digital twin while eliminating the need for a centralised storage system to hold data.

### 2.2. Blockchain and Applications

Blockchain is an essential technology of the current era that stores transactional records in a block-like structure. The block storage consists of databases, often referred to as Distributed Ledger Technology (DLT), chained to their adjacent blocks, forming a secure chain of blocks. The whole process is conducted through a peer-to-peer (P2P) network, where every node comprises a copy of the ledger. The transaction between these nodes are recorded in the time-stamped log in the shared ledger, and the encryption feature ensures security of data transmission on the blockchain.

Blockchain can be suitable for DT following the ten-step decision path suggested by [18] to determine which type of blockchain technology to use. The requirements state that Step 1, considering the issue of scalability and storing large amounts of data on the blockchain, can be slow, and the cost is also high because of the transaction fees. To solve this issue, blockchain should be integrated with an off-chain database. Step 2–4, if there are multiple parties with trust issues involved, then the blockchain system is considered as a solution. Step 5–6, blockchain is a solution if transaction rules remain largely unchanged, and rules governing the system access differ with different participants depending on their roles and characteristics. Step 7, blockchain is of great significance with immutable logs, and it is good for auditing and security. Step 8–10, the type of blockchain to use is determined by the digital twin system, based on the consensus amongst the participants.

Combining blockchain with DT will ensure trust between participants of the DT. The participants are registered as authorised users in the blockchain. The DT generates relevant information and stores historical data in the blockchain, which will be further analysed, as shown in Figure 2. Blockchain ensures data security and will help predict and evaluate critical events about the DT [19].

## 3. Related Works

Digital twin is still an emerging technology, with limited availability of blockchain-based solutions and literature. In this section, we discuss existing work and blockchain-based solutions for the DT.

The prototype implementation EtherTwin, developed by [5], shows how numerous implementation challenges are associated with the fully decentralised sharing of data, enabling the management of DT information and components with their associated information. The authors proposed a decentralised owner-centric sharing model, with an access control model to overcome the aspect of data confidentiality and integrity on the digital twin components and life cycle requirements to tackle the need for decentralised sharing of data on the digital twin to avoid a single point of trust. The authors [5] discussed that prototypical implementations were not considered, that the approaches were theoretical, and they were not implemented.Swarm was considered in this implementation; Swarm is restricted to one update per second. The timely sharing of DT data are restricted to one update per second, which is insufficient to support the frequent updates of sensor data occurring multiple times per second. However, this aspect was not considered in the scope of this research.

Similar work was undertaken by [1]. Using the Ethereum blockchain, the authors proposed a blockchain-based creation process of the digital twin to achieve immutability, accessibility, traceability, trust, and security for data transactions. They used smart contracts to initiate participants involved in the creation of a digital twin to govern and track transactions. An integrated digital twin and blockchain framework is developed and tested by [3] to achieve data communication and traceability. The framework is a virtual representation of a prefabricated brick as a project use-case, where the virtual positioning data are transmitted to the digital twin in near real time and recorded on the blockchain. The result of their framework shows that data transactions are traceable. This scheme uses a centralised storage or third party, and this can be a limitation to a single point of failure or vulnerability to system attacks.

Digital twin data management can be complicated to manage from the perspectives of data access, data storage, data sharing, and data authenticity [9]. The author proposed a blockchain-based method for the data management of digital twin products, using a peer-to-peer network constructed to enhance data sharing efficiency among participants. The results indicate that data can be stored in blocks and accessed with verification. Efficient data sharing is facilitated through a peer-to-peer network, while data authenticity is ensured through traceability, thereby preventing data overwriting. The authors [19] discuss the integration of DTs and blockchain for the industrial internet of things (IIoT) to address the issues of data management and security. They discussed identifying outstanding challenges faced by the IIoT and not implementing them. The author focused on the adoption of a lightweight, scalable, and quantum-immune blockchain-based solution for IIoT.

The combination of IIoT and private blockchain-driven digital twin was proposed by [20] for configuring intelligent manufacturing systems (IMSs). The IMSs architecture is based on the IIoT, which has a centralised architecture that limits the capacity to support security and has poor flexibility to handle manufacturing disturbances. The evaluation was based on throughput and latency of Ethereum and Fabric, and the outcome showed that Fabric could achieve higher throughput and lower latency than Ethereum. The evaluation of time latency is required if blockchain and cloud storage are applied to DT data sharing; thus, the time latency has not been evaluated for the combination of HyperledgerFabric and cloud storage.

Ref. [21] proposed a blockchain-based framework for securing big digital twin data (BDTD). The idea is based on large amounts of digital twin data generated in the life cycle of equipment used in the digital twin-enabled applications. However, the lack of security leads to data sharing issues. The use of performance-based contracts is to consider the overall life cycle of equipment rather than the production costs. The challenge of performance evaluation, financials, and accountability has been limited, so [7] proposes the connection of the digital twin with blockchain-based smart contracts to handle performance-based digital payment to achieve transparency and trust for performance evaluation.

The aforementioned state-of-the-art focused mainly on data sharing and data management, and the use of blockchain in DT. However, some schemes have limitations as a result of the use of Swarm to achieve timely data sharing—the use of centralised data storage, which can lead to a single point of attack, while some schemes are focused on the security of data management of IIoT without been implemented.

To address such limitations, our proposed system uses a decentralised approach to data management and sharing, leveraging blockchain technology, smart contracts, and IPFS. By using a decentralised approach, we can avoid the risk of a single point of failure and improve the security and reliability of the system. The proposed system offers an innovative solution that can address the limitations of existing approaches and potentially provide a more secure and efficient method for managing and sharing data in the context of digital twins.

Table 1 summarises the existing work for blockchain and DT.

## 4. Proposed Blockchain-Based System

This section describes the proposed blockchain-based approach for secure communication between DTs. It provides an overview of the DApps entities, explaining the DT and blockchain components. The architecture of the system is represented in Figure 3, which provides an overview of the DT and blockchain sharing approach.

This work utilises a permissioned blockchain implemented through Ethereum, with decentralised applications (dApps) such as Hyperledger and Besu [22]. Our work adopts the dApp Ganache. Ganache is a personal blockchain for rapid Ethereum- and Corda-distributed application development. This will allow a permissioned blockchain where authorised participants are permitted to access the blockchain.

The bi-directional communication interface is connected to the sensor, enabling the collection of real-time data from the real-world asset (Conveyor Belt). These data can be in the form of specifications or documents, which are then sent and stored on the IPFS. The smart contract will be utilised to create stakeholders and grant access to the DT’s information, as well as to control access to this information from the IPFS.

The components of the proposed system are presented in Figure 3. Web3.js is utilised to send transactions signed with the private key to the smart contracts on the Ethereum blockchain. Truffle serves as a development environment that includes a local blockchain development server that launches automatically. JSONRPC is employed to collect blockchain data from client programs, parse the retrieved data, store it in the database, and display the data.

The DT and the physical asset are connected through a bi-directional communication interface. To enable data sharing, the DApp is introduced, and the DT data are stored on the IPFS. The front-end DApp utilises Web3, JSONRPC, and Truffle interfaces to connect the participants in the DT to the smart contract and IPFS servers. This interaction with the smart contract ensures secure and trusted information sharing among them.

Our algorithm ensures the integrity and traceability of documents within the digital twin system. Ownership of digital twins is tracked through Ethereum addresses, and participants interact with the system by calling smart contract functions. Access to functions is restricted based on Ethereum addresses using a modifier. When a participant uploads a document, it is stored in IPFS, and its hash value is recorded in the smart contract on the Ethereum blockchain. This allows for the verification of document integrity by comparing the stored hash with the actual document. Participants can request approval by providing the hash value, ensuring ownership and authorisation. The algorithm’s process is illustrated in Figure 4, showcasing the sequence of the file upload process.

### 4.1. Advantages of Using Ethereum Blockchain

Ethereum is an open-source blockchain-based platform that allows computer applications to run on top of it. There is no need to build your own blockchain, as it allows developers to program their own smart contracts using the Solidity language. DApps are developed for these applications that are running on the blockchain to communicate in the blockchain network with each other.

The Ethereum blockchain uses the Proof of Work (PoW)-based consensus protocol to run applications within the blockchain. In this protocol, transactions are verified in the blockchain network through mining. Miners make transactions and prove their work by accomplishing a computationally complex task, which leads to heavy power consumption. The Ethereum network is moving to the Proof of Stake (PoS)-based consensus protocol. Here, the miner having the highest stake of the total cryptocurrency will be able to verify transactions on the blockchain. The PoS-based protocol is considered to solve the power consumption problem and is less computationally expensive for its function.

Ethereum is a decentralised blockchain platform that enables the creation of decentralised applications (dApps) and smart contracts. Here are some of the advantages of using the Ethereum blockchain:Decentralisation: Ethereum is a decentralised platform that operates on a network of nodes, which makes it resistant to censorship and a single point of failure.Smart Contracts: Ethereum allows the creation of self-executing smart contracts that automate the execution of agreements between parties, without the need for intermediaries.Transparency: The Ethereum blockchain is transparent and public, which allows for easy auditing and the verification of transactions and smart contracts.Security: Ethereum uses cryptography to secure its network, which ensures that transactions and smart contracts cannot be altered or tampered with.Interoperability: Ethereum’s open-source nature and support for programming languages such as Solidity and Vyper allows it to integrate with other blockchain platforms and applications.Tokenisation: Ethereum allows for the creation and management of tokens, which can be used for various purposes, such as crowdfunding, loyalty programs, and governance.Community: Ethereum has a large and active developer community that is constantly working on improving the platform and building new applications.Ethereum provides a robust and secure platform for building decentralised applications and smart contracts, with the potential to revolutionise various industries by enabling new business models and creating more transparent and efficient systems.

### 4.2. System Architecture

This section describes the components of the proposed blockchain-based system for DT. The following are the components of the proposed system:

#### 4.2.1. Digital Twin Stakeholders

It is important to have a clear and well-defined set of stakeholders for any digital twin system, as this helps ensure that the system is properly managed and that the right people have access to the right information and functions.

In the proposed system, the stakeholders are delivery manager, manufacturer, maintainer, testing manager, quality manager, and process manager. Each of these stakeholders has specific roles and responsibilities within the digital twin system, and they are authorised to perform certain functions in the smart contract.

The process manager, who is also known as the smart contract owner, is responsible for initiating the creation process of DTs. This role is particularly important, as it ensures that the digital twin system is created and managed in a way that is consistent with the overall goals and objectives of the organisation.

The delivery manager is responsible for the release of the asset for deployment when the DT is ready for deployment. This role is critical, as it ensures that the digital twin system is properly deployed and integrated into the larger organisational infrastructure.

The manufacturer, maintainer, testing manager, and quality manager also have important roles and responsibilities within the digital twin system. They may be responsible for creating, maintaining, testing, and ensuring the quality of the digital twin system, depending on the specific requirements and goals of the organisation.

Having a clear set of stakeholders and roles within the digital twin system can help ensure that the system is properly managed and that the right people have access to the right information and functions. By leveraging the power of smart contracts and IPFS storage, the proposed system can provide a secure and efficient way to manage the creation and deployment of digital twin systems.

#### 4.2.2. Ethereum Smart Contract

Introducing an Ethereum smart contract to manage the creation of participants in a digital twin system can provide a number of benefits, including increased transparency, immutability, and security.

Solidity is the most popular language for writing smart contracts on the Ethereum platform, and it provides a number of features for defining and implementing contract logic. Compiling the smart contract with a node.js environment can help simplify the development and testing process by providing access to a range of libraries and tools for interacting with the Ethereum network.

Using a local server such as Ganache can also be useful for testing and development purposes, as it provides a lightweight and configurable environment for running and testing smart contracts. This can help developers iterate more quickly and identify potential issues before deploying the contract to the main Ethereum network.

Introducing an Ethereum smart contract to manage the creation of participants in a digital twin system can be a powerful tool for improving security, transparency, and efficiency. However, it is important to ensure that the smart contract is well-designed and thoroughly tested to avoid potential vulnerabilities or issues.

Accessing the digital twin information from the IPFS servers will be made accessible by the smart contract that deals with the IPFS hash, as shown in Listing 1.

**Listing 1.** Smart contract code.
pragma solidity 0.5.0;


contract DStorage {

    string public name = "DStorage";

    uint public fileCount = 0;

    mapping(uint => File) public files;


    struct File {

        uint fileId;

        string fileHash;

        uint fileSize;

        string fileType;

        string fileName;

        string fileDescription;

        uint uploadTime;

        address payable uploader;

    }


    event FileUploaded(

        uint fileId,

        string fileHash,

        uint fileSize,

        string fileType,

        string fileName,

        string fileDescription,

        uint uploadTime,

        address payable uploader

    );


    // ERC-20 Token implementation

    //The unique token can be used to ensure that only

      authorized participants can access the data, by

      providing a means of verifying the identity of the

      user attempting to access the information.


    string public constant symbol = "DST";

    string public constant name = "DStorage Token";

    uint8 public constant decimals = 18;

    uint256 public totalSupply;


    mapping(address => uint256) public balanceOf;


    event Transfer(address indexed from, address indexed to, uint256 value);


    constructor(uint256 initialSupply) public {

        totalSupply = initialSupply;

        balanceOf[msg.sender] = initialSupply;

    }


    function transfer(address _to, uint256 _value) public returns (bool success) {

        require(balanceOf[msg.sender] >= _value);

        balanceOf[msg.sender] -= _value;

        balanceOf[_to] += _value;

        emit Transfer(msg.sender, _to, _value);

        return true;

    }


    // File storage functions


    function uploadFile(

        string memory _fileHash,

        uint _fileSize,

        string memory _fileType,

        string memory _fileName,

        string memory _fileDescription

    ) public {

        require(bytes(_fileHash).length > 0, "File hash is required");

        require(bytes(_fileType).length > 0, "File type is required");

        require(bytes(_fileDescription).length > 0, "File description is required");

        require(bytes(_fileName).length > 0, "File name is required");

        require(msg.sender != address(0), "Uploader address is required");

        require(_fileSize > 0, "File size must be greater than zero");


        fileCount++;

        files[fileCount] = File(

            fileCount,

            _fileHash,

            _fileSize,

            _fileType,

            _fileName,

            _fileDescription,

            now,

            msg.sender

        );


        emit FileUploaded(

            fileCount,

            _fileHash,

            _fileSize,

            _fileType,

            _fileName,

            _fileDescription,

            now,

            msg.sender

        );

    }

}



#### 4.2.3. Blockchain Decentralised Storage

Using IPFS as an off-chain storage system for the digital twin system can offer a number of benefits, such as decentralisation, immutability, and accessibility.

By using IPFS to store relevant information for the digital twin, the system can benefit from a decentralised storage infrastructure that provides increased reliability and accessibility. Unlike traditional centralised storage systems, IPFS uses a distributed network of nodes to store and share data, which helps ensure that the information is always available, even if some nodes go offline.

Every file uploaded to the IPFS has a generated unique hash of the IPFS that guarantees integrity and is stored in the smart contract [1]. Information stored on the IPFS is private, as transactions are not visible on the public blockchain.

Additionally, information stored on the IPFS network is immutable, meaning that once it is stored, it cannot be altered or deleted. This can help ensure the integrity and authenticity of the information, which is particularly important for sensitive data related to the digital twin.

Furthermore, transactions on IPFS are recorded instantly and usually free of charge, which can help reduce the transaction costs associated with on-chain storage systems such as Ethereum. This can help make the digital twin system more affordable and accessible to a wider range of users.

Using IPFS as an off-chain storage system can provide a powerful and reliable way to store information for the digital twin system, while also leveraging the benefits of decentralisation, immutability, and accessibility.

### 4.3. Implementation

We used IPFS, as it is considered to be more efficient than BitTorrent and Git file storage systems. IPFS provides a high throughput, content-addressed block storage model, which ensures the security of transactions [23].

The blockchain DApp prototype was developed on the Ethereum blockchain platform. The platform supports the Turing complete smart contract programming language Solidity and has a large developer community, resulting in advanced development tools and vulnerability scanners. The system has a user interface that simplifies the interactions of the DT involved in creating twins and uploading data. For a trustless interaction with the blockchain, it is implemented using the web page application JavaScript framework Node.JS, with the Ganache local blockchain server only needed to serve static assets. The module metamask wallet is used for managing the user’s blockchain account, providing access to the user’s public and private key.

To implement a DApp, a blockchain network was built using Ganache Ethereum in a local environment, and Truffle was used as a framework in compiling and deploying smart contracts. The DApp accesses the smart contract by communicating with the Ganache Ethereum via JSONRPC through Web3js, an API for the Ethereum JavaScript.

The DT can be owned by several participants; when the DT is deployed, the new owner is registered. The Ethereum address of the DT owners can be tracked and traced using the events available log. The owner is responsible for the creation process of the DTs, and each of the participants has an Ethereum blockchain address and participates by calling functions in the smart contract at certain times. Access is restricted on who can execute a function call by the modifier; this is done by using the Ethereum address of the participant.

The registered owner creates the smart contract as well as the document for upload, which includes the document’s name and the document IPFS hash. This process can be done by any registered participant to upload a new version of the document and also view documents uploaded to the IPFS. Figure 4 illustrates the sequence for the file upload process. The uploaded document on the IPFS stores the hash value in the contract, and approval is requested by the participant by providing the hash.

#### 4.3.1. Approving Document Version

The function requesting approval for the document version will execute only when the participant requesting the approval is registered and the document hash matches the IPFS hash stored in the smart contract.

Listing 2 provides a clear outline of the process for registered participants to request approval for updating a document version on IPFS. The algorithm outlines the following steps:

Check that the participants requesting approval are registered and the document hash matches the IPFS hash stored in the smart contract.

If the conditions are met, change the state of the contract from Created to WaitForApproversSignature and the state of the Participant to SubmittedForApproval.

Notify the transaction with an event, requesting validation of the document from at least two-thirds of all approvers.

If at least two-thirds of the approvers validate the document, update the IPFS hash stored in the smart contract to the new document hash.

Change the state of the contract from WaitForApproversSignature to Updated and the state of the developer to UpdatedDocument.

Notify the transaction with an event indicating that the document has been successfully updated.

If the conditions in step 1 are not met, then the transaction aborts and returns to its initial state. This helps ensure the integrity and security of the digital twin system by preventing unauthorised modifications or submissions of false documents.

Listing 2 provides a clear and concise outline of the process for requesting approval for updating a document version on IPFS, and the conditions and steps outlined help ensure the security and reliability of the digital twin system.

**Listing 2.** Pseudocode for approving the document version.
// Define the structure for a participant

struct Participant {

   bool registered;

   bool approved;

}


// Contract state

enum ContractState {

  Created,

  WaitForApproversSignature,

  Reverted

}



# Define contract states


class ContractState:

  Created = "Created"

  WaitForApproversSignature = "WaitForApproversSignature"



# Define participant states


class ParticipantState:

  ReadyToSubmit = "ReadyToSubmit"

  SubmittedForApproval = "SubmittedForApproval"



# List of registered participants


RegisteredParticipants = []


def validateParticipant(participantAddress, IPFSHash):

   contractState = ContractState.Created

   participantState = ParticipantState.ReadyToSubmit


   if participantAddress in RegisteredParticipants:

        restrictAccessToParticipant(participantAddress)


        if IPFSHash:

            contractState = ContractState.WaitForApproversSignature

            participantState = ParticipantState.SubmittedForApproval

            createValidationMessage("Requesting validation from all approvers")

        else:

            revertContractState("IPFS hash is not provided")

   else:

        revertContractState("Participant is not registered")


def restrictAccessToParticipant(participantAddress):

   # Restrict access to only the specified participant

   pass


def createValidationMessage(message):

   # Create a validation message with the given content

   pass


def revertContractState(error):

   # Revert contract state and show error message

   pass


#### 4.3.2. Consent for Uploaded Documents

The requesting participant must be registered and the hash provided as input must be the same as the IPFS hash stored in the contract.

Listing 3 outlines the process for all approvers to provide their consent for a document version uploaded on IPFS. The algorithm includes the following steps:Check that the requesting participant is registered and the hash provided as input is the same as the IPFS hash stored in the contract.If the conditions are met, change the state of the contract to SignatureProvided and the state of the developer to ApprovalProvided.For each approver, if they provide successful approval, change their state to ApprovalSuccess.If all approvers provide successful approval, change the state of the contract to Updated and the state of the participant to UpdatedDocument.Notify the transaction with an event indicating that the document has been successfully updated.If any approver fails to provide approval, change the state of the contract to SignatureDenied and the state of the participant to ApprovalNotProvided.Change the state of all approvers who failed to provide approval to ApprovalFailed.

Listing 3 provides a clear and concise outline of the process for all approvers to provide their consent for a document version uploaded on IPFS. The algorithm includes checks to ensure the requesting developer is registered and the hash matches the IPFS hash stored in the contract, and the algorithm handles scenarios when successful approval is provided or not provided by the approvers. These conditions and steps help ensure the security and reliability of the digital twin system.

**Listing 3.** Pseudocode for consent for uploaded documents.
// Define the structure for a participant

struct Participant {

   bool registered;

   bool approved;

}


// Contract state

enum ContractState {

  Created,

  WaitForApproversSignature,

  Reverted

}



# Define contract states


class ContractState:

  WaitingForApproversSignature = "WaitingForApproversSignature"

  SignatureProvided = "SignatureProvided"

  SignatureDenied = "SignatureDenied"



# Define participant states


class ParticipantState:

  ReadyToSubmit = "ReadyToSubmit"

  ApprovalProvided = "ApprovalProvided"

  ApprovalNotProvided = "ApprovalNotProvided"



# Define approver states


class ApproversState:

  WaitingToSign = "WaitingToSign"

  ApprovalSuccess = "ApprovalSuccess"

  ApprovalFailed = "ApprovalFailed"


def validateParticipant(participantAddress, documentHash, IPFSHash):

   contractState = ContractState.WaitingForApproversSignature

   participantState = ParticipantState.ReadyToSubmit

   approversState = ApproversState.WaitingToSign


   if participantAddress in RegisteredParticipantSet:

        restrictAccessToParticipant(participantAddress)


        if documentHash == IPFSHash:

            contractState = ContractState.SignatureProvided

            participantState = ParticipantState.ApprovalProvided

            approversState = ApproversState.ApprovalSuccess

            createValidationMessage("Request ID granted")

        else:

            contractState = ContractState.SignatureDenied

            participantState = ParticipantState.ApprovalNotProvided

            approversState = ApproversState.ApprovalFailed

            createValidationMessage("Document version approval failed")

   else:

        revertContractState("Participant is not registered")


def restrictAccessToParticipant(participantAddress):

   # Restrict access to only the specified participant

   pass


def createValidationMessage(message):

   # Create a validation message with the given content

   pass


def revertContractState(error):

   # Revert contract state and show error message

   pass


#### 4.3.3. Processing Registration Requests

With a successful request, the state of the contract changes to “NewRegRequested” and that of the new registration state changes to “NewRegistrationRequested”, and a notification message is broadcasted to all registered participants to approve of the new registration.

In Listing 4 the function request NewRegistration takes in three parameters: name, email, and address, which represent the name, email, and address of the participant requesting new registration in the network. The function first checks if the participant is not already registered, and then sets the state of the registration to WaitToRegister.

Next, a new registration entry is created with the provided details, and the approval count is initialised to 0. The state of the new registration is set to NewRegistrationRequested, and the new registration entry is added to the registrations mapping with the requesting participant’s address as the key.

Finally, a message is broadcasted to all registered approvers and participants about the new registration request, and the function returns true.

**Listing 4.** Pseudocode for processing registration requests.
// Define the structure for a participant

struct Participant {

   bool registered;

   bool approved;

}


// Contract state

enum ContractState {

  Created,

  WaitForApproversSignature,

  Reverted

}



# Define contract states


class ContractState:

  SignatureProvided = "SignatureProvided"

  NewRegRequested = "NewRegRequested"

  Reverted = "Reverted"



# Define registered states


class RegisteredState:

  HaitToRegister = "HaitToRegister"

  NewRegisteredRequested = "NewRegisteredRequested"


def processNewEntrant(ethereumAddressOfNewEntrant):

   contractState = ContractState.SignatureProvided

   registeredState = RegisteredState.HaitToRegister


   if isAlreadyRegistered(ethereumAddressOfNewEntrant):

        revertContractState("New entrant is already registered")

   else:

        contractState = ContractState.NewRegRequested

        registeredState = RegisteredState.NewRegisteredRequested

        approversState = "ApprovalFailed"

        createNotification("Grant permission for new registrations")


def isAlreadyRegistered(ethereumAddress):

   # Check if the new entrant is already registered

   # Return True if registered, False otherwise

   pass


def revertContractState(error):

   # Revert contract state and show error message

   pass


def createNotification(message):

   # Create a notification message for granting permission

   pass


## 5. Case Scenario

This case scenario is motivated by an industrial size conveyor as shown in Figure 5 and Figure 6, which demonstrate how this conveyor will be used with the proposed blockchain solution in practice by creating a digital twin. To demonstrate the effectiveness of the proposed system, a case scenario about a production line of a smartphone, where a conveyor belt is used to carry different parts of the smartphone production, is presented. The main goal for the smartphone production is to gather all data available about the conveyor belt and to monitor and analyse the data.

This case study is intended to verify and manage the DT over time, which generates a compliance statement that includes a big file size of data, which can be shared immutably and traceably through the blockchain network. The DT historical data can be stored without the issue of overwriting and shared in the blockchain network with limited storage. A digital twin is created for each smartphone being produced, which includes information such as:Component Information: Data related to the components used in smartphone assembly, such as the supplier, batch number, manufacturing date, and quality certifications.Machine Sensor Data: Real-time data from machines and equipment involved in the assembly and testing processes, including temperature, pressure, vibration, and energy consumption.Production Metrics: Data on production cycle time, yield rates, defect rates, and other performance indicators to monitor the efficiency and quality of the manufacturing process.Quality Assurance: Inspection results, including images, videos, and test reports, generated during quality control procedures to identify any defects or deviations.Supply Chain Data: Information about the origin, transportation, and storage conditions of raw materials and components used in smartphone production, ensuring transparency and traceability.

To perceive these data, a combination of technologies is utilised, including IoT sensors, machine vision systems, RFID tags, and data-logging mechanisms. These technologies enable real-time data capture, monitoring, and analysis throughout the production line.

The collected data serves multiple purposes, aimed at improving the overall efficiency, quality, and trustworthiness of the smartphone production process. Some specific objectives and benefits include:Traceability: By capturing and recording the component information and supply chain data, the system ensures end-to-end traceability of each smartphone unit, enabling effective quality control, recall management, and counterfeit prevention.Predictive Maintenance: By continuously monitoring machine sensor data, potential faults and maintenance needs can be detected in advance, allowing proactive maintenance to prevent unplanned downtime and production delays.Process Optimisation: Analysis of production metrics and machine data facilitates identifying bottlenecks, optimising workflows, and improving overall production efficiency, resulting in reduced costs and increased output.Quality Assurance: The inspection results and associated data help identify defects early in the process, enabling timely corrective actions, reducing rework, and ensuring the delivery of high-quality smartphones to end-users.

The data are collected using sensors and IoT devices, and the digital twin is updated in real time. The data from the digital twin are stored in a distributed file system such as IPFS, ensuring its immutability and authenticity. The data are then anchored onto a blockchain network, providing a tamper-proof record of the data’s origin and provenance.

Decentralised applications (DApps) will be built on top of the blockchain to provide various functionalities such as real-time monitoring, analytics, and reporting. These DApps can be utilised to monitor the production line, track the progress of each smartphone throughout the assembly process, and identify any real-time issues or defects.

Smart contracts will be deployed on the blockchain to automate various processes such as quality control and payments. For example, when a smartphone passes a quality control check, a smart contract could be used to automatically trigger the payment to the supplier of the components. Access to the data and the blockchain is managed through cryptographic keys, ensuring that only authorised parties can access the data and execute transactions on the blockchain.

By leveraging the benefits of digital twin and blockchain technology, it is possible to create a more secure, efficient, and transparent production line for smartphones. The system can help improve quality control, reduce the risk of counterfeit products, and provide valuable insights for optimising the production line and creating new business models.

## 6. Evaluations and Analysis

This section presents a comparison of the performance between the proposed system and existing methods [1,5]. The proposed system is evaluated in terms of security and performance, with performance being determined by the deployment cost of the smart contracts. The smart contract was tested and simulated using the Remix IDE. The Remix IDE provides features for creating, deploying, testing, and debugging smart contracts. Different Ethereum accounts were utilised to test the smart contract through the Remix IDE and Ganache. Users with specific roles are granted permission to perform certain actions related to the state of the digital twin in the smart contract.

### 6.1. Security Analysis

To ensure security, smart contract implementation should be tested to avoid attacks on the smart contract. In a live network, extreme measures are taken in a decentralised blockchain network because the deployed smart contract cannot be modified or upgraded. Applications of the smart contract cannot be modified even by the developers after deployment [24]. Due to the immutable nature of the smart contract, hackers are unable to make changes to the smart contract, which gives an added security feature. Testing smart contracts for vulnerabilities has become important before deployment and should be tested with a wide range of test cases for security reasons. Smart contracts, blockchain nodes, wallets, and consensus mechanisms are the common elements that can suffer from vulnerabilities on the blockchain.

Using a combination of a time-stamp and unique token can be an effective way to ensure the confidentiality of information data in a DT system. A time-stamp can help prevent replay attacks by ensuring that any data that are being accessed are current and have not been tampered with or replayed from a previous interaction. The smart contract is written in Solidity and the token adopted the Solidity standard, which is the ERC-20, as shown in Listing 1. As a fungible token, the unique token can be used to ensure that only authorised participants can access the data by providing a means of verifying the identity of the user attempting to access the information.

In addition, this combination can help prevent MITM attacks, where an attacker attempts to intercept and manipulate data as they are being transmitted between two parties. By including a time-stamp and unique token in the data being transmitted, any attempt to manipulate or intercept the data will likely result in a failed verification process and prevent unauthorised access.

Using a time-stamp and unique token can be a useful security measure in a DT system, as it can help ensure confidentiality and prevent both replay and MITM attacks. However, it is important to ensure that the implementation of these measures is done correctly to avoid any potential vulnerabilities in the system.

Reentrancy is indeed a well-known vulnerability in smart contracts that can be exploited by attackers to drain funds or cause other unintended behaviour. It occurs when a contract calls an external contract, and the external contract calls back into the original contract before the initial call has finished executing.

One way to prevent reentrancy attacks is to use the “checks–effects–interactions” pattern, where all checks and state changes are performed before any external interactions are made. Additionally, setting the gas limit manually for each transaction can also prevent reentrancy attacks.

Regarding the issue with the proposed system and the fallback function, it is important to note that the absence of a fallback function does not necessarily mean that the system is immune to reentrancy attacks. While a fallback function can indeed create a potential avenue for an attacker to exploit, reentrancy can still occur through other means such as external contract calls.

To prevent reentrancy attacks in the proposed system, it is important to carefully design and implement the smart contracts to follow best practices such as the “checks–effects–interactions” pattern and setting gas limits manually. Additionally, comprehensive testing and auditing of the code can help identify and mitigate potential vulnerabilities.

### 6.2. Performance

In this section, we present the comparative performance between the proposed system and the existing method. In our prototype implementation, we use the Ethereum blockchain, the cryptocurrency for this blockchain is ether (ETH), and the fee is calculated based on the transaction and execution cost (gas amount). The gas amount required depends on the smart contract implementation and the number of transactions needed to execute the smart contract.

#### 6.2.1. Registration

The smart contract execution for the registration of participants in a blockchain environment is displayed in Figure 7 for transaction and execution costs. The transaction cost decreases by 10,896 gas, and the execution cost increases by 11,720 gas when compared with the existing method [1].

The proposed system has 59,681 gas for transaction gas cost and 59,681 gas for execution gas cost, while the existing method has 70,577 gas for transaction gas cost and 47,961 gas for execution gas cost. The transaction cost of the existing method is higher than the proposed system, and the execution cost is lower. This is because storing in an array incurs higher costs, and storing in an array involves storing the length of the element. Changing the value of storage space from zero to non-zero costs much higher for the first time in any type of storage [1].

On the other hand, the proposed system uses IPFS for storage, which is a more efficient and cost-effective solution. IPFS allows for instant transactions without having to wait for network confirmations, and transactions done off-chain are usually free, resulting in lower transaction costs. Additionally, IPFS offers greater privacy and security, as data transfers are not visible in the public blockchain.

By using a more efficient and cost-effective storage solution such as IPFS, the proposed system is able to achieve lower transaction costs while still maintaining high levels of security and privacy.

#### 6.2.2. Deployment

The smart contract execution for the deployment of contracts in the Ethereum blockchain is displayed in Figure 8 and Figure 9 for the gas transaction and ether costs. The initial deployment gas transaction used 24,464 for the proposed system with the ether (Eth) cost at 0.00489 and the initial deployment gas of 14,548 with ether cost at 0.14548 for the existing method.

The proposed system made use of an IPFS for storage, while the existing method made use of Swarm for storage. The use of IPFS for storage in the proposed system offers several advantages over the existing method that uses Swarm for storage. IPFS is faster and more efficient, as transactions are recorded instantly without having to wait for network confirmations, which can result in significant delays. Additionally, transactions completed on off-chain are usually free, making it a cost-effective solution for data storage.

Another advantage of IPFS is that it offers greater privacy and security, as data transfers are not visible in the public blockchain. This is particularly important for applications that involve sensitive or confidential information.

However, it is worth noting that using a Content Identifier (CID v1) rather than a (CID v0) can result in higher gas consumption. Gas consumption refers to the amount of computational resources needed to execute a transaction on the blockchain. Therefore, it is important to carefully consider the trade-offs between the benefits of using IPFS and the associated gas costs when designing and implementing a blockchain-based digital twin system.

The use of IPFS for storage in the proposed system offers several advantages over existing methods, including faster transaction speeds, lower costs, and greater privacy and security. By carefully managing gas consumption and other performance factors, it is possible to create a highly effective and efficient system that can benefit a wide range of industries and applications.

#### 6.2.3. IPFS Storage

The results for the IPFS file storage are displayed in Table 2. The result shows a difference of 0.000002 Eth between a 14 kB, 252 KB, and 2 MB document. Documents that are 14 KB and 252 KB have 0.007406 Eth respectively, while a 2 MB document has 0.007408 Eth, as this demonstrates the low-cost transactions of an IPFS storage.

IPFS uses a content-addressed system, which means that the content of a file is used to generate a unique hash that is used to retrieve the file. When a file is added to IPFS, it is broken up into smaller chunks, each of which is addressed by its own unique hash. This means that when a file is retrieved from IPFS, only the specific chunks needed to reconstruct the file are retrieved, rather than the entire file.

This content-addressed system allows for very efficient storage and retrieval of data, as only the required portions of a file need to be accessed. In addition, IPFS uses a peer-to-peer network for distribution and storage, which further reduces the costs associated with storing and accessing data.

The low transaction costs associated with IPFS storage make it an attractive option for a variety of use-cases, particularly those involving large amounts of data or frequent access to stored data. However, it is important to note that the actual costs associated with IPFS transactions can vary depending on a variety of factors, including network congestion, gas prices, and other external factors.

## 7. Discussion

This section discusses the results of the evaluation, potential threats on the proposed system, and the performance evaluation of the proposed system.

### 7.1. Security Threat

Our solution employes blockchain to secure the creation of DTs to ensure the traceability and immutability of DTs’ data. One of the most important features of the blockchain is its data integrity. Due to their low cost, off-chain transactions are gaining popularity among large participants. All transactions have a hash, which makes transactions on the off-chain safe from a man in the middle attack (MITM), thereby making them immutable. If a public blockchain is used, the confidentiality of on-chain metadata becomes a concern [5]. Due to a lack of trust between participants, a consensus is required to be strict. The fact that users are required to pay fees to participate in the network is a problem that is associated with a public blockchain [25]. Information distributed across the public blockchain network and made available for every participant poses a challenge to information privacy. Both the Ethereum blockchain and IPFS can be set up as private networks to ensure control and data confidentiality.

Confidentiality becomes a concern, if s public blockchain is used, as data are not encrypted in on-chain storage. Participants should therefore avoid storing confidential information in on-chain storage due to security risks. To ensure data confidentiality, an off-chain storage is used rather than an on-chain storage, as [5] indicated that storing confidential information in on-chain storage is not considered safe. The proposed system is set up as a private network with the use of Ethereum blockchain and IPFS to ensure data confidentiality.

The proposed system uses IPFS to store and share data among the participants and also ensures accountability for the DTs by restriction due to the smart contract function. These functions can be amended to meet the specific needs of different industries. The application of blockchain is resilient to several attacks and security vulnerabilities for the creation process of DTs.

### 7.2. Performance

The evaluation results demonstrate the potential of our proposed system in achieving low-cost transactions and also in supporting timely data sharing. As [5] indicated, in on-chain, 50 and 100 transactions/second are supported in the private Ethereum blockchains, and this means that more than 4 million interactions are possible per day. However, in off-chain, multiple transactions are shared per second, while Swarm is currently restricted to one update per second. Data shared in off-chain are secured, no data are lost, and data can be shared in a timely fashion.

Furthermore, the use of digital twins ensures that the data being shared are accurate and up-to-date, reducing the risk of errors and improving the efficiency of the system. Additionally, the use of smart contracts can help to automate various processes, reducing the need for manual intervention and improving the speed and accuracy of transactions.

The combination of digital twin and blockchain technology also provides a high level of security, with data being encrypted and stored in a decentralised manner, making it difficult for unauthorised parties to access or tamper with the data.

The proposed system has the potential to significantly improve the efficiency, transparency, and security of data sharing and transactions, while also reducing costs and improving the speed of transactions. As the technology continues to evolve and improve, it is expected that the potential benefits of this system will become even more pronounced, making it a valuable tool for businesses and organisations in a wide range of industries.

## 8. Conclusions

This paper has explored the underlying issues hindering information management in DT. Blockchain-based digital twin sharing is designed to secure the information of the DT during sharing. The performance evaluation of our proposed system shows that it is secure and achieves performance improvement when compared with other methods. The comparison of results with existing methods showed that the proposed system consumed less gas. During registration, the transaction cost decreases by 10,896 gas, and the execution cost increases by 11,720 gas when compared with the existing method. During deployment, the initial deployment gas transaction used 24,464 for the proposed system with the ether (Eth) cost at 0.00489 and the initial deployment gas 14,548 with an ether cost at 0.14548 for the existing method.

One of the key advantages of the proposed system is its ability to ensure data confidentiality. By using off-chain storage instead of on-chain storage, the system can securely store confidential information without the risk of it being accessed by unauthorised parties. The use of a private network with the Ethereum blockchain and IPFS further enhances data confidentiality, making the system suitable for use in a wide range of industries and sectors.

In future work, the proposed system can be extended to other areas such as healthcare, business, transportation, and more. By leveraging the benefits of digital twin technology and blockchain, it is possible to create more efficient, secure, and transparent systems that can benefit a wide range of stakeholders.

It is crucial for the system architects and developers to carefully analyse the scalability requirements, evaluate potential bottlenecks, and design the system accordingly to ensure it can handle the increasing demands and data volumes associated with a large number of digital twins.

Overall, the proposed system has the potential to address the challenges of information management in digital twin technology, providing a secure and efficient way to share and manage information. As the technology continues to evolve and improve, it is expected that the potential benefits of this system will become even more pronounced, making it an increasingly valuable tool for businesses and organisations in a wide range of industries. The proposed system has the potential to address the challenges of information management in digital twin technology, providing a secure and efficient way to share and manage information. This is particularly significant in the context of Industry 5.0, where the integration of advanced technologies, such as the Internet of Things (IoT), artificial intelligence (AI), and automation, are revolutionising the manufacturing and industrial sectors.

## Figures and Tables

**Figure 1 sensors-23-06641-f001:**
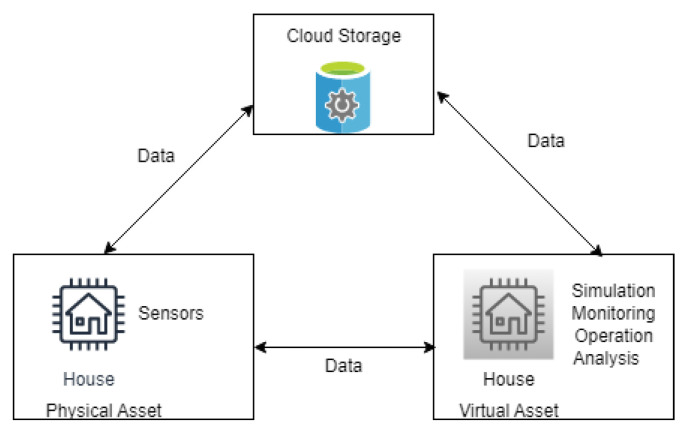
Three principal components of digital twins.

**Figure 2 sensors-23-06641-f002:**
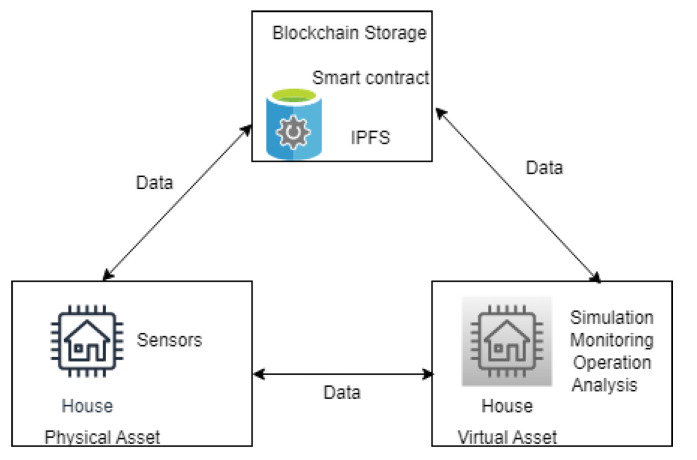
Digital twins with blockchain-based data storage.

**Figure 3 sensors-23-06641-f003:**
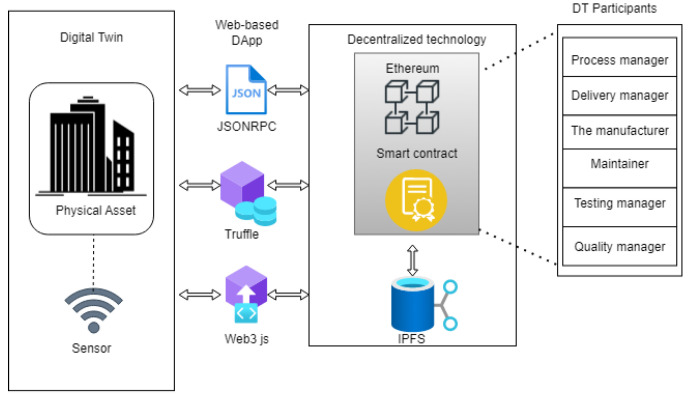
The system architecture showing how the DT interacts with the blockchain.

**Figure 4 sensors-23-06641-f004:**
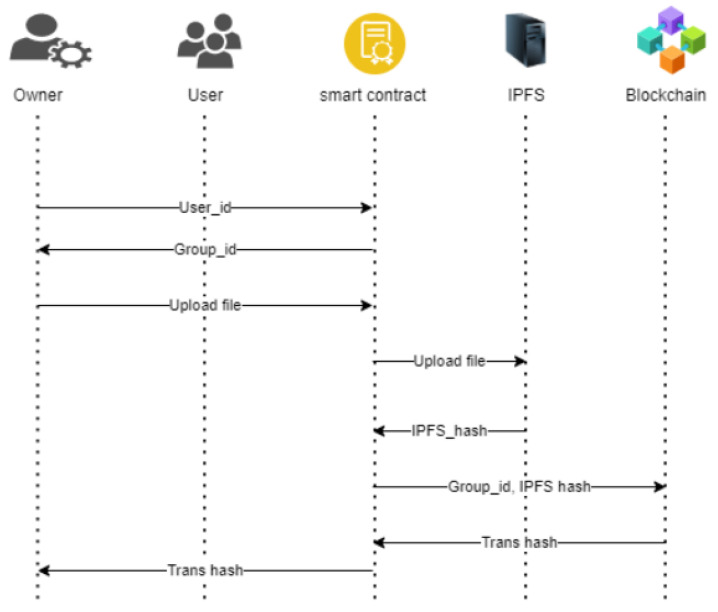
Sequence demonstration showing all the interactions between the participants of the smart contract for the file upload process.

**Figure 5 sensors-23-06641-f005:**
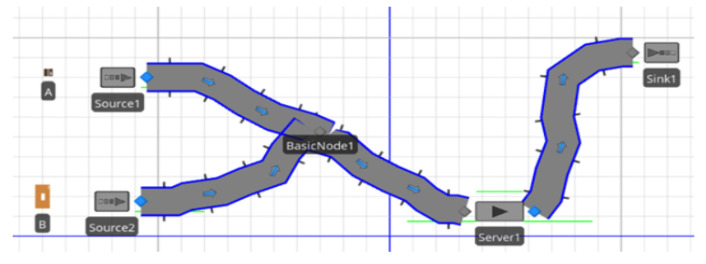
Two-dimensional view of an industrial-sized conveyor.

**Figure 6 sensors-23-06641-f006:**
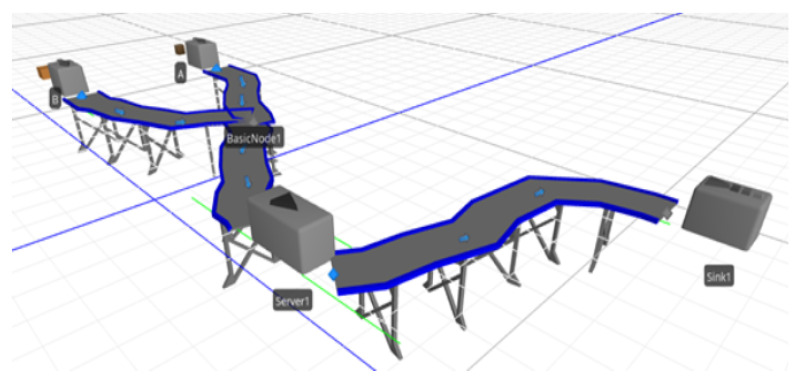
Three-dimensional view of an industrial-sized conveyor.

**Figure 7 sensors-23-06641-f007:**
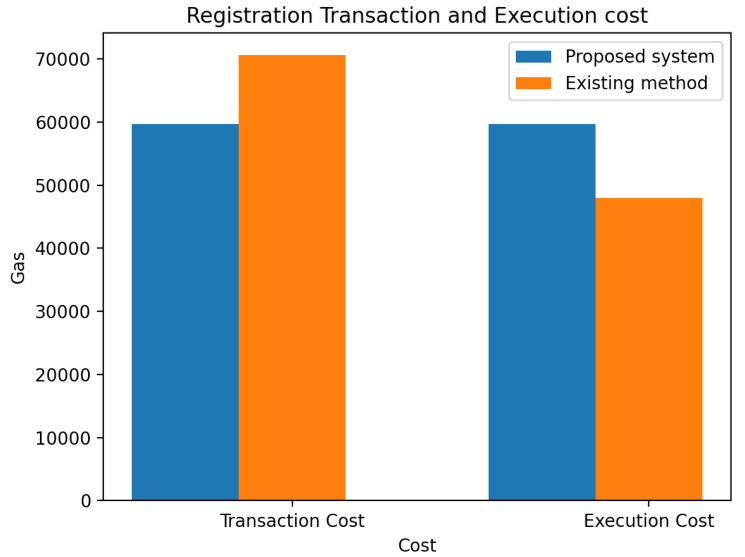
Comparison of experimental results of proposed system and existing system on registra- tion transaction cost and execution cost.

**Figure 8 sensors-23-06641-f008:**
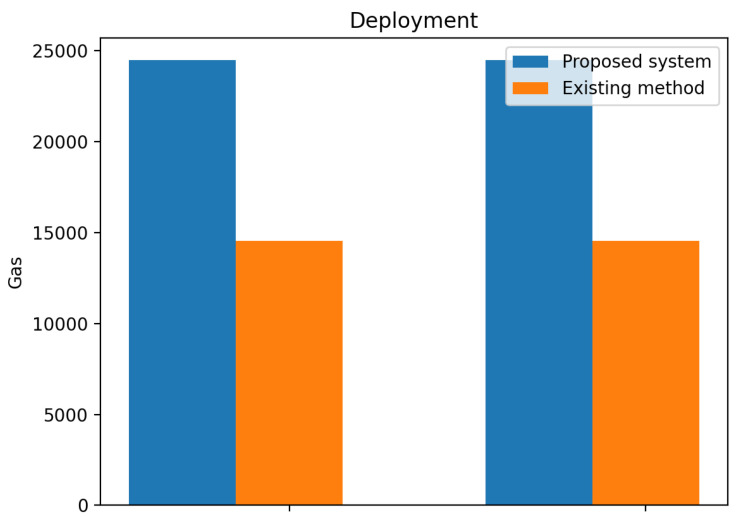
Comparison of experimental results of proposed system and existing system on deployment transaction cost.

**Figure 9 sensors-23-06641-f009:**
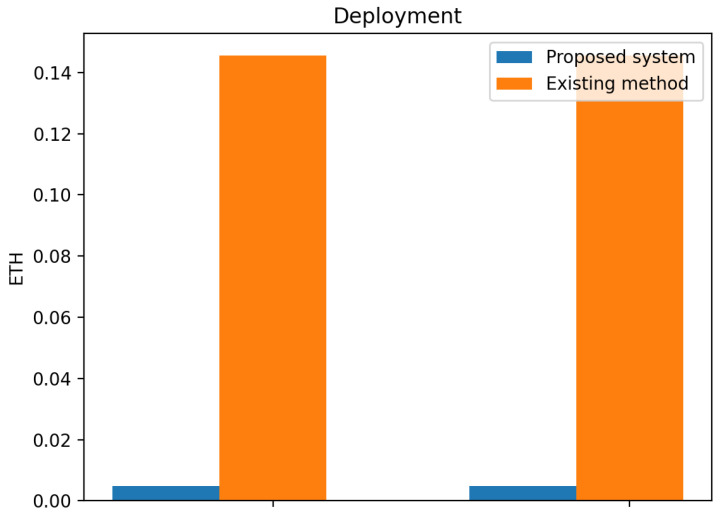
Comparison of experimental results of proposed system and existing system on deployment ether cost.

**Table 1 sensors-23-06641-t001:** Comparison of blockchain and digital twin-related works.

Related Works	Year	Purpose	Blockchain	Blockchain Type	Data Storage	Implementation	Test Results
[5]	2021	Secure data sharing	Ethereum	Unknown	Off-chain (swarm)	Fully	Ethereum and Swarm, and evaluated the latency and cost
[1]	2020	Secure creation process for digital twin	Ethereum	Public	Off-chain (IPFS)	Partially	Provides security and evaluates the cost
[3]	2021	Accountable information sharing	Azure platform: blockchain	Private	Cloud Storage	Unknown	Digital twin can generate compliance statements in real time
[9]	2020	Data management of digital twin	Unknown	Unknown	On-chain	Unknown	Case study to show effectiveness of the proposed method
[19]	2021	Data Management	DAG-based blockchain	None	On-chain	None	None
[20]	2020	Configuration of digital twin manufacturing cell	Fabric	IIoT and Private	Off-chain (Edge)	Partially	Fabric could achieve higher throughput and lower latency than Ethereum
[21]	2021	Secure sharing of big data	Fabric	Unknown	Cloud Storage	Unknown	Shows that it has better performance, which is a common price strategy
[7]	2021	Performance cost	Ethereum	Unkown	Unkown	Fully	Performance evaluation through smart contracts

**Table 2 sensors-23-06641-t002:** Storage cost of IPFS files.

Size	ETH
14 KB	0.007406
252 KB	0.007406
2 MB	0.007408

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
