# Peer review of "Enabling Trust and Security in Digital Twin Management: A Blockchain-Based Approach with Ethereum and IPFS"

_sensors, 2023, doi:10.3390/s23146641_

Round 1
Reviewer 1 Report
The authors propose a blockchain-based solution for digital twins in the Ethereum ecosystem. The solution uses smart contracts for information management and IPFS as a database for documents. To illustrate their proposal, the authors propose a real case study. The results presented seem promising. However, I would like to make some suggestions and comments:
- In Figure 4, the authors make a diagram to explain their proposal. You can see how they store the documents in the IPFS, and what they store in the blockchain is the hash of this document. In this way, it is possible to check whether the document has been altered, in addition to its traceability. However, reading the text of the article, this is only explained at the end of the article. It would be good to explain how the algorithm works in the introduction (briefly) and in section 4, as a reader unfamiliar with blockchain may have gaps when trying to understand how it works.
- In section 5, the authors propose a case scenario where they create a digital twin for every smartphone produced. Would this scenario have scalability issues in a real-world environment?
- In Section 6.1, the authors claim to have performed a security analysis of their smart contracts. How did they do this? Did they use one of the classic tools such as Mythril or Slither, or did they run a series of unit tests?
-On page 19, the authors talk about a 'unique token'. Is this token fungible or non-fungible? What kind of smart contract was used to create it? Did it use the solidity standards?
- There are some typos: pag 10, 'Figure ??'. Caption figure 8 and 9, 'double dot ..', etc
The article is well written and within the scope of the journal. I only recommend a major revision according to my comments and suggestions.
Author Response
Original Manuscript ID: sensors-2499611
Original Article Title: Enabling Trust and Security in Digital Twin Management: A Blockchain based Approach with Ethereum and IPFS
Date: 17th July, 2023
To: Sensors Editor
Re: Response to reviewers
Dear Editor,
We would like to express our sincere gratitude to all reviewers for their insightful comments on the manuscript. We are also grateful to the Editor and reviewers for the opportunity to submit the revised version of the manuscript. In the revised manuscript the enhancements of the contents (addition, modification, and deletion) made to incorporate reviewer’s comments have been explicitly highlighted with yellow and a short comment:
In the following, we answer reviewers’ comments individually. For each comment, enhancements made to the original manuscript are explicitly mentioned with section, page (of the revised manuscript); where reviewers’ have only asked the further explanation, we are providing detail/justification in this response letter.
For this revision we have uploaded the following files:
- our point-by-point response to the comments (response to reviewers).
- an updated manuscript with yellow highlighting indicating changes.
- a clean updated manuscript without highlights (PDF main document).
We sincerely believe that valuable suggestions and comments provided by the reviewers have significantly improved the overall quality of this manuscript aligned with the quality of Sensors. This revision has helped to further enhance the research rigour, contents, structural outline, and readability of the manuscript.
Best regards,
Zeeshan PERVEZ, PhD
SFHEA, SMIEEE, ACM Distinguished Speaker, EPSRC Peer Review College
Professor in Computer Science
School of Computing, Engineering & Physical Sciences
University of the West of Scotland
High Street, Paisley, PA1 2BE
United Kingdom
Email: zeeshan.pervez@uws.ac.uk
Tel: +44 141 848 3183
Reviewer 1:
The authors propose a blockchain-based solution for digital twins in the Ethereum ecosystem. The solution uses smart contracts for information management and IPFS as a database for documents. To illustrate their proposal, the authors propose a real case study. The results presented seem promising. However, I would like to make some suggestions and comments:
- In Figure 4, the authors make a diagram to explain their proposal. You can see how they store the documents in the IPFS, and what they store in the blockchain is the hash of this document. In this way, it is possible to check whether the document has been altered, in addition to its traceability. However, reading the text of the article, this is only explained at the end of the article. It would be good to explain how the algorithm works in the introduction (briefly) and in section 4, as a reader unfamiliar with blockchain may have gaps when trying to understand how it works.
Author response: Thank you for your valuable feedback.
Author action: Brief introduction about how the algorithm works, has been added to the Introduction Section 1 on page 3, and Section 4 on page 8 of the paper as advised.
- In section 5, the authors propose a case scenario where they create a digital twin for every smartphone produced. Would this scenario have scalability issues in a real-world environment?
Author response: Thank you for your insightful comment on the manuscript.
Author action: The scalability of creating a digital twin for every smartphone produced in a real-world environment can indeed pose challenges. While digital twins can offer valuable insights and benefits, managing many digital twins at scale may present scalability issues. This Limitation has been added to Section 8 of the paper.
- In Section 6.1, the authors claim to have performed a security analysis of their smart contracts. How did they do this? Did they use one of the classic tools such as Mythril or Slither, or did they run a series of unit tests?
Author response: Thank you for your comment.
Author action: Security analysis was performed on smart contracts, by running series of unit tests with Remix IDE. This was explained in the first paragraph of Section 6 page 20, which I will highlight on my updated manuscript.
- On page 19, the authors talk about a 'unique token'. Is this token fungible or non-fungible? What kind of smart contract was used to create it? Did it use the solidity standards?
Author response: Thank you for your valuable feedback.
Author action: The Smart contract was written in Solidity and the token adopted the Solidity standard, which is the ERC-20, and it is a fungible token as shown in smart contract Listing 1 subsubsection 4.2.2.
- There are some typos: page 10, 'Figure ??'. Caption figure 8 and 9, 'double dot ..', etc
Author response: Thank you for your feedback.
Author action: The page 10 figure typo is corrected, along with the caption of Figure 8 and 9 double dot typo.
Reviewer 2 Report
This manuscript proposed a blockchain based approach with ethereum and IPFS to enable trust and security in digital twin. Combining blockchain with DT is important and interesting in the related fields. The proposed system in this work has good innovation and effectiveness. However, in section 5 of case scenario, the case of a production line of a smartphone is provided, which is too general. I suggest providing more specific information, such as what data is collected, how it is perceived, what its purpose is, and so on.
Author Response
Original Manuscript ID: sensors-2499611
Original Article Title: Enabling Trust and Security in Digital Twin Management: A Blockchain based Approach with Ethereum and IPFS
Date: 17th July, 2023
To: Sensors Editor
Re: Response to reviewers
Dear Editor,
We would like to express our sincere gratitude to all reviewers for their insightful comments on the manuscript. We are also grateful to the Editor and reviewers for the opportunity to submit the revised version of the manuscript. In the revised manuscript the enhancements of the contents (addition, modification, and deletion) made to incorporate reviewer’s comments have been explicitly highlighted with yellow and a short comment:
In the following, we answer reviewers’ comments individually. For each comment, enhancements made to the original manuscript are explicitly mentioned with section, page (of the revised manuscript); where reviewers’ have only asked the further explanation, we are providing detail/justification in this response letter.
For this revision we have uploaded the following files:
- our point-by-point response to the comments (response to reviewers).
- an updated manuscript with yellow highlighting indicating changes.
- a clean updated manuscript without highlights (PDF main document).
We sincerely believe that valuable suggestions and comments provided by the reviewers have significantly improved the overall quality of this manuscript aligned with the quality of Sensors. This revision has helped to further enhance the research rigour, contents, structural outline, and readability of the manuscript.
Best regards,
Zeeshan PERVEZ, PhD
SFHEA, SMIEEE, ACM Distinguished Speaker, EPSRC Peer Review College
Professor in Computer Science
School of Computing, Engineering & Physical Sciences
University of the West of Scotland
High Street, Paisley, PA1 2BE
United Kingdom
Email: zeeshan.pervez@uws.ac.uk
Tel: +44 141 848 3183
Reviewer 2:
- This manuscript proposed a blockchain based approach with Ethereum and IPFS to enable trust and security in digital twin. Combining blockchain with DT is important and interesting in the related fields. The proposed system in this work has good innovation and effectiveness. However, in section 5 of case scenario, the case of a production line of a smartphone is provided, which is too general. I suggest providing more specific information, such as what data is collected, how it is perceived, what its purpose is, and so on.
Author response: Thank you for highlighting this.
Author action: The type of data to be collected, how it is perceived, the purpose and benefits of data collection has been provided in Section 5 as advised.
Round 2
Reviewer 1 Report
The authors have responded satisfactorily to all my considerations. I have no further comments.